# Relationship between cigarette smoking and blood pressure in adults in Nepal: A population-based cross-sectional study

**Renqiao Lan**[1], **Max K. Bulsara**[2], **Prakash Dev Pant**[3], **Hilary Jane Wallace**[1,4]*

**1** School of Medicine, The University of Notre Dame Australia, Fremantle, Western Australia, Australia,
**2** Institute for Health Research, The University of Notre Dame Australia, Fremantle, Western Australia,
Australia, **3** Monitoring and Evaluation Consultant, Kathmandu, Nepal, **4** School of Population and Global
Health, The University of Western Australia, Crawley, Western Australia, Australia

* hilary.wallace@uwa.edu.au

org/10.1371/journal.pgph.0000045

University of Health Sciences, BANGLADESH

**Data Availability Statement:** Nepal Demographic
and Health Survey 2016 [Dataset]. NPPR7HFL.SAV
(household member recode); NPIR7HFL.SAV
(individual recode); NPMR7HFL.SAV (men's

## Abstract

Smoking and hypertension are two major risk factors for cardiovascular disease, the leading
cause of death in Nepal. The relationship between cigarette smoking and blood pressure
(BP) in Nepal is unclear. This study analysed the data from the 2016 Nepal Demographic
and Health Survey to explore the differences in systolic BP (SBP) and diastolic BP (DBP)
between current daily cigarette smokers and non-smokers in Nepali adults aged 18 to 49
years. A total of 5518 women and 3420 men with valid BP measurements were included.
Age, body mass index, wealth quintile (socio-economic status) and agricultural occupation
(proxy for physical activity) were included as potential confounders in multivariable linear
regression analysis. Women smokers were found to have significantly lower SBP (mean dif-
ference 2.8 mm, 95% CI 0.7–4.8 mm) and DBP (mean difference 2.2 mm, 95% CI 0.9–3.6
mm) than non-smokers after adjustment. There were no significant differences in BP
between smokers and non-smokers in males, either before or after adjustment. The lower
BP in female cigarette smokers in Nepal may be explained by the physiological effect of
daily cigarette smoking *per se* in women, or unmeasured confounders associated with a tra-
ditional lifestyle that may lower BP (for example, diet and physical activity). In this nationally
representative survey, daily cigarette smoking was not associated with increased BP in
males or females in Nepal.

## Introduction

Cardiovascular disease is the leading cause of death globally [1] and in Nepal [2]. Both cigarette
smoking and hypertension (high blood pressure) are well-established risk factors for cardio-
vascular disease and are thought to act synergistically on disease development [3–6]. Accord-
ing to the Nepal Burden of Disease 2017 report, high blood pressure and smoking are the top
two risk factors for death and are responsible for 14% and 13% of all deaths in Nepal respec-
tively [2]. The aetiology of primary hypertension is complex and lifestyle risk factors such as
obesity, physical inactivity, excessive alcohol consumption and high salt intake are proposed to
be strongly and independently associated with its development [7–9].

recode). Rockville, Maryland: Ministry of Health, New ERA, and ICF [Producers]. ICF [Distributor], 2017. Available from https://dhsprogram.com/data/dataset/Nepal_Standard-DHS_2016.cfm?flag=0.

**Funding:** This research was performed in part by a student (RL) and staff (HW, MB) of The University of Notre Dame Australia, Fremantle. RL received a short-term mobility scholarship from the Endeavour Leadership Program (ELP) of the Australian Government to undertake research training with PP and HW in Nepal. The University of Notre Dame and the ELP were not involved in the study design, data collection and analysis, decision to publish, or preparation of the manuscript.

**Competing interests:** The authors declare that there are no competing interests regarding the publication of this article.

The understanding of the role of cigarette smoking in hypertension development continues to be refined. The hemodynamic effects of cigarette smoking are mediated primarily by nicotine [10], which can increase blood pressure (BP) acutely and temporarily via stimulation of the sympathetic nervous system [10–12]. However, with long-term exposure nicotine may have different effects [13]. For example, it is hypothesized that the nicotine metabolite, cotinine, may decrease BP via its vasodilatory effect [13]. Nicotine may also decrease BP via lowering body weight secondary to its effects of appetite suppression or increasing metabolism [14].

Epidemiological studies on the relationship between smoking and BP have produced mixed results. Some studies have found a positive association between current smoking and hypertension [15–17], including in an urban Nepali population [18]. By contrast, BP has also been found to be the same or lower in many groups of smokers compared to non-smokers [17, 19–23].

The nationally representative 2016 Nepal Demographic and Health Survey (NDHS) report found that 17% of women and 23% of men (aged 15 years and over) were hypertensive, and the cigarette smoking rate was 5.5% among women and 26.9% in men (aged 15–49 years) [24]. Cigarette smoking is the most common form of tobacco smoking by men and women in Nepal [24]. Although demographic factors and overweight/obesity were found to be associated with hypertension in adults in this survey [25–27], the relationship between cigarette smoking and BP needs further study. Given the high burden of cardiovascular disease, it is important to have a better understanding of the relationship between its two major risk factors in Nepal's unique sociodemographic context and add to the evidence available from this specific population to inform mechanistic studies. The aim of this study is to explore the relationship between cigarette smoking and BP (systolic and diastolic) in the Nepali adult population aged 18–49 years, using data from the 2016 Nepal Demographic and Health Survey.

## Methods

### Study design, setting, and participants

Data were obtained from the 2016 Nepal Demographic and Health Survey (NDHS), a nationally representative cross-sectional household survey, funded by the US Agency for International Development (USAID) [24]. The survey was conducted from June 19, 2016, to January 31, 2017, and the sampling frame was a modified version of the Nepal Central Bureau of Statistics 2011 National Population and Housing Census [24].

Households were selected in two stages in rural areas and three stages in urban areas [24]. In urban areas, wards (smallest units of local government in Nepal) were the primary sampling units (PSU), from which one enumeration area (EA) was selected. Households were subsequently selected from EAs [24]. In rural areas, wards were the PSU from which households were selected directly [24]. Only households containing a woman aged 15–49 years the night before survey administration were eligible for interview. All women aged 15–49 years who were permanent residents of the selected household or visitors who stayed the night in the household the night before the survey were eligible to be interviewed. All men aged 15–49 years from every second household who were permanent residents of the selected household or visitors who stayed the night in the households the night before the survey were eligible to be interviewed. Full details of the NDHS sampling design are discussed elsewhere [24]. Blood pressure measurements were recorded in women and men only in the subsample of households selected for the male survey [24]. Daily cigarette smoking was recorded for all participants aged 15–49 years who were interviewed.

Participants included in this study were men and women aged 18–49 years who were interviewed and with a valid BP measurement. Participants taking BP lowering medication were

excluded from the study. Of the 12862 total women aged 15–49 years in the survey, 6452 women had BP measured. After inclusion and exclusion criteria were applied (62 women with technically invalid BP readings, 796 women under 18 years, and 92 women on BP lowering medication) there were 5518 women for analysis. Of the 4063 total men aged 15–49 in the survey, 4040 men had BP measured. After the exclusions were applied (22 men with technically invalid BP readings, 543 men under 18 years, and 71 men on BP lowering medication) there were 3420 men for analysis.

## Variables

Systolic BP (SBP) and diastolic BP (DBP) were the primary outcome variables. Blood pressure was measured three times in each participant at a minimum of five-minute intervals using the UA-767F/FAC blood pressure monitor (A&D Medical, Japan). The first measurement was discarded and the average of the second and third measurements was recorded as the final reading and recorded as a continuous variable (mm Hg) according to the standard DHS biomarker collection protocol [24].

Current cigarette smokers were defined as those who smoked cigarettes daily (manufactured or hand-rolled). Cigarette smokers were further categorized according to number of cigarettes smoked per day (up to 9, 10 or more). For men, this number was the average daily number of cigarettes in the past week, and for women, the number of cigarettes in the last 24 hours.

Demographic variables used to describe the participants were age (years), body mass index (BMI) (weight (kg)/height (m)$^2$), education (Y/N), economic status (poorest wealth quintile; Y/N), social group (marginalised group, non-marginalised group), agricultural occupation (Y/N), place of residence (urban, rural), Province (1–7) and ecological zone (mountains, hills, Terai [plains]). For wealth quintile we used the NDHS household wealth index, derived from detailed information on dwelling and household characteristics, access to a variety of consumer goods and services, and assets [24]. Classification of participants as marginalised or non-marginalised was based on an ethnic grouping which is reflective of the social hierarchy in Nepal [28]. The marginalised group comprised Terai Dalit, Hill Dalit, Hill Janajati, Terai Janajati, Muslim and other Terai castes. Participants not in these groups were classified as non-marginalised.

In exploring the association between cigarette smoking and BP, age, body mass index, socioeconomic status and physical activity were considered potential confounders. Age was classified into the following sub-groups: 18–24 years, 25–34 years, 35–44 years, 45–49 years. Body mass index (BMI) was categorised according to the World Health Organization general population BMI classification into underweight (<18.5 kg/m$^2$), normal (18.5 to 24.9 kg/m$^2$), overweight (25.0 to 29.9 kg/m$^2$) and obese ($\geq$30.0 kg/m$^2$). Socioeconomic status [29] was assessed through economic status (wealth quintile) and social group (marginalised ethnic group: Y/N). Physical activity was accounted for, in part, through the proxy variable of agricultural occupation (Y/N), with agricultural occupation representing higher physical activity [30, 31].

## Statistical analysis

The data were analysed using IBM SPSS Ver 26.0 software (IBM Corp., Armonk, N.Y., USA). Data were weighted using sampling weights in accordance with DHS guidelines [32]. All analyses used the Complex Sample Analysis method to account for the multi-stage sample design [32]. Data from men and women were analysed separately. There were no missing data.

The relationship between smoking and BP was assessed with linear regression. The dependent variables in linear regression were the continuous variables SBP and DBP. The potential confounders were treated as categorical variables: age, BMI, wealth quintile, social group, and agricultural work. SBP and DBP were adjusted for age (through linear regression) after stratification into the four age groups (18–24 years, 25–34 years, 35–44 years, 45–49 years). All potential confounders which showed a significant association with BP (in either sex) in age-adjusted linear regression were included in the final multivariable linear regression models. The outcomes are presented as mean SBP and DBP with 95% confidence intervals.

Tests for interactions were also carried out, fitting a smoking X BMI interaction term in the models for men and women, with cigarette smoking fitted as a 2-category variable (Y/N) and BMI as a 4-category variable (underweight, normal, overweight, and obese).

## Power analysis

Using the OpenEpi2 Sample Size calculator for power analysis (comparing two means) [33] and data from the 2016 NDHS (sample size, smoking prevalence and standard deviation), there was 80% statistical power to detect a 2.5 mm Hg mean difference in systolic BP between cigarette smokers and non-smokers in women, and a 1.8 mm Hg mean difference in men.

## Ethics approval

The 2016 NDHS survey protocol was approved by the Nepal Health Research Council (NHRC) and the ICF Institutional Review Board prior to administration. Written informed consent was obtained from individual respondents prior to the interviews during the NDHS data collection. Access to the NDHS 2016 dataset for this project was granted by the DHS Program before the study was carried out. The study was also approved by the Human Research Ethics Committee of the University of Notre Dame Australia, Fremantle (Ref. 2020-066F).

## Results

The characteristics of smokers in our sample (Table 1) showed several differences between men and women and to non-smokers. A smaller proportion of women (4.8%) smoked cigarettes daily than men (18.9%), and the same proportion (27%) of women and men smokers were moderate to heavy smokers (10 cigarettes or more per day [17]). While both men and women cigarette smokers had lower BMI than non-smokers, the mean difference in BMI was larger in women (2.1 units vs. 0.8 units in women and men respectively). A higher proportion of women smokers (41.8%) than men smokers (25.5%) were in the poorest wealth quintile, and a much higher proportion of women smokers (86.2%) had no formal education compared to men who smoked (16.8%) or to women who did not smoke (34.7%). Women smokers were, on average, older (mean 39.7 years) than non-smokers (mean 30.5 years) and men who smoked (mean 33.9 years). Women smokers were more often engaged in agricultural work (64.9%) than non-smokers (46.7%) and men who smoked (30.8%). The proportion of women smokers and men smokers in marginalised social groups was the same (68.8%).

After adjustment for age, mean SBP and DBP were strongly associated with BMI category in both men and women, with significantly higher mean BP in overweight (4–6 mm) and obese (7–10 mm), and lower BP in underweight (3–6 mm), compared to the normal BMI group (Table 2). Men and women in the richest wealth quintile had significantly higher mean BP than those in middle wealth quintile (except for SBP in women), but this was a smaller effect (approximately 2–3 mm) than BMI. Men who were not engaged in agricultural work, but not women, had significantly higher BP than those who were, by approximately 2 mm. Mean BP was similar in the marginalised and non-marginalised social groups.

**Table 1. Characteristics of participants by smoking status.**

| Characteristic | | Current smoker[1] | | Current smoker, cigarettes/day[2] (manufactured or hand-rolled) | |
|---|---|---|---|---|---|
| | | No | Yes | Up to 9 | 10+ |
| **Men** (n = 3420) | | | | | |
| n | | 2772 | 647 | 471 | 176 |
| % | | 81.1 | 18.9 | 13.8 | 5.1 |
| BMI, kg/m$^2$ (mean ± SEM) | | 22.3 ± 0.09 | 21.6 ± 0.15 | 21.8 ± 0.18 | 21.1 ± 0.26 |
| BMI, kg/m$^2$ [age-adjusted] (mean ± SEM) | | 22.4 ± 0.10 | 21.6 ± 0.14 | 21.8 ± 0.16 | 21.0 ± 0.26 |
| Age, years (mean ± SEM) | | 31.0 ± 0.24 | 33.9 ± 0.43 | 33.3 ± 0.49 | 35.5 ± 1.00 |
| Poorest quintile (%[3]) (n = 518) | | 12.8 | 25.5 | 21.7 | 35.7 |
| No education (%[3]) (n = 379) | | 9.7 | 16.8 | 16.7 | 17.0 |
| Social group (%[3]) | Marginalised group [Terai Dalit, Hill Dalit, Hill Janajati, Terai Janajati, Muslim, other Terai Caste] (n = 2246) | 65.0 | 68.8 | 69.9 | 65.8 |
| | Non-marginalised group [Hill Brahmin, Hill Chhetri, Terai Brahmin, Terai Chhetri, Newar] (n = 1174) | 35.0 | 31.2 | 30.1 | 34.2 |
| Agricultural occupation (%[3]) (proxy for physical activity) | Yes (n = 984) | 28.3 | 30.8 | 27.7 | 38.9 |
| | No (n = 2436) | 71.7 | 69.2 | 72.3 | 61.1 |
| Place of residence (%[3]) | Urban (n = 2228) | 64.4 | 68.5 | 71.7 | 59.7 |
| | Rural (n = 1192) | 35.6 | 31.5 | 28.3 | 40.3 |
| Province (%[3]) | Province 1 (n = 578) | 17.1 | 16.0 | 17.8 | 11.4 |
| | Province 2 (n = 675) | 21.7 | 11.1 | 13.1 | 6.0 |
| | Province 3 (n = 880) | 24.0 | 33.1 | 28.7 | 44.9 |
| | Province 4 (n = 302) | 8.9 | 8.8 | 9.4 | 7.0 |
| | Province 5 (n = 551) | 16.9 | 12.8 | 15.5 | 5.6 |
| | Province 6 (n = 167) | 4.5 | 6.3 | 5.6 | 8.2 |
| | Province 7 (n = 267) | 6.9 | 11.8 | 9.9 | 16.7 |
| Ecological zone (%[3]) | Mountain (n = 207) | 5.3 | 9.1 | 6.9 | 14.9 |
| | Hill (n = 1511) | 43.5 | 47.1 | 40.7 | 64.3 |
| | Terai (n = 1702) | 51.2 | 43.8 | 52.4 | 20.8 |
| **Women** (n = 5518) | | | | | |
| n | | 5251 | 267 | 196 | 71 |
| % | | 95.2 | 4.8 | 3.6 | 1.3 |
| BMI, kg/m$^2$ (mean ± SEM) | | 22.6 ± 0.11 | 21.3 ± 0.23 | 21.2 ± 0.25 | 21.5 ± 0.54 |
| BMI, kg/m$^2$ [age-adjusted] (mean ± SEM) | | 22.8 ± 0.12 | 20.7 ± 0.24 | 20.6 ± 0.26 | 20.8 ± 0.52 |
| Age, years (mean ± SEM) | | 30.5 ± 0.13 | 39.7 ± 0.58 | 39.2 ± 0.62 | 41.2 ± 1.14 |
| Poorest quintile (%[3]) (n = 920) | | 15.4 | 41.8 | 41.7 | 42.2 |
| No education (%[3]) (n = 2053) | | 34.7 | 86.2 | 83.9 | 92.7 |
| Social group (%[3]) | Marginalised group [Terai Dalit, Hill Dalit, Hill Janajati, Terai Janajati, Muslim, other Terai Caste] (n = 3496) | 63.1 | 68.8 | 71.1 | 62.4 |
| | Non-marginalised group [Hill Brahmin, Hill Chhetri, Terai Brahmin, Terai Chhetri, Newar] (n = 2022) | 36.9 | 31.2 | 28.9 | 37.6 |
| Agricultural work (%[3]) (proxy for physical activity) | Yes (n = 2625) | 46.7 | 64.9 | 66.3 | 61.0 |
| | No (n = 2893) | 53.3 | 35.1 | 33.7 | 39.0 |
| Place of residence (%[3]) | Urban (n = 3460) | 62.7 | 62.1 | 61.1 | 65.1 |
| | Rural (n = 2058) | 37.3 | 37.9 | 38.9 | 34.9 |
| Province (%[3]) | Province 1 (n = 931) | 17.3 | 7.9 | 8.9 | 5.2 |
| | Province 2 (n = 1119) | 21.0 | 6.5 | 6.1 | 7.4 |
| | Province 3 (n = 1205) | 21.3 | 32.3 | 27.0 | 47.0 |
| | Province 4 (n = 543) | 9.8 | 9.8 | 11.0 | 6.1 |
| | Province 5 (n = 932) | 17.0 | 14.0 | 17.0 | 5.8 |
| | Province 6 (n = 306) | 5.1 | 13.5 | 12.6 | 16.0 |
| | Province 7 (n = 482) | 8.4 | 16.0 | 17.3 | 12.4 |
| Ecological zone (%[3]) | Mountain (n = 330) | 5.8 | 10.3 | 11.5 | 6.9 |
| | Hill (n = 2431) | 43.3 | 60.0 | 56.7 | 69.1 |
| | Terai (n = 2757) | 51.0 | 29.7 | 31.8 | 24.0 |

[1] Smokes cigarettes daily.

[2] Men: average daily number of cigarettes in past week; Women: number of cigarettes in last 24 hours.

[3] Column percent.

**Table 2. Age-adjusted mean SBP and DBP by confounding variables.**

| Characteristic | | Mean SBP, mm Hg | | Mean DBP, mm Hg | |
|---|---|---|---|---|---|
| | | Men | Women | Men | Women |
| BMI, kg/m2 | <18.5 | 112.0# | 105.4# | 75.2# | 72.6# |
| | ≥18.5–25* | 118.0 | 109.3 | 78.9 | 75.6 |
| | ≥25–30 | 124.3# | 113.6# | 84.9# | 80.1# |
| | ≥30 | 127.4# | 119.0# | 85.8# | 83.6# |
| Wealth quintile | Poorest | 118.9 | 110.3 | 79.8 | 76.2 |
| | Poorer | 118.6 | 111.2¥ | 79.2 | 77.2 |
| | Middle* | 117.0 | 109.9 | 78.5 | 76.1 |
| | Richer | 118.1 | 109.0 | 79.3 | 75.6 |
| | Richest | 120.1# | 110.8 | 81.1‡ | 77.8¥ |
| Social group | Marginalised | 118.4 | 110.6 | 79.4 | 76.7 |
| | Non-marginalised* | 118.5 | 109.6 | 80.2 | 76.3 |
| Agricultural work (proxy for physical activity) | Yes* | 117.4 | 110.5 | 78.3 | 76.3 |
| | No | 118.9¥ | 110.0 | 80.3# | 76.8 |

*Reference category.

¥ p<0.05.

‡ p<0.005.

# p<0.001.

After age-adjustment, women smokers overall had significantly lower mean SBP (mean difference 3.9 mm; 95% CI 1.7–6.0 mm) and lower mean DBP (mean difference 3.4 mm, 95% CI 1.9–4.8 mm) than non-smokers (Table 3). There were no significant differences in BP between smokers and non-smokers in men, either before or after age-adjustment.

**Table 3. Unadjusted and age-adjusted mean SBP and DBP by smoking status.**

| BP, mmHg (95% CI) | | Current smoker[1] | | Current smoker, cigarettes/day[2] (manufactured or hand-rolled) | |
|---|---|---|---|---|---|
| | | No* | Yes | Up to 9 | 10+ |
| Men (n = 3420) | | | | | |
| SBP | Unadjusted | 117.3 (116.4–118.2) | 118.8 (117.0–120.5) | 118.8 | 118.7 |
| | Adjusted | 118.5 (117.6–119.5) | 118.9 (117.1–120.6) | 119.1 | 118.3 |
| DBP | Unadjusted | 78.8 (78.0–79.5) | 79.3 (78.0–80.6) | 79.5 | 78.8 |
| | Adjusted | 79.8 (79.0–80.5) | 79.2 (78.0–80.4) | 79.6 | 78.3 |
| Women (n = 5518) | | | | | |
| SBP | Unadjusted | 108.3 (107.6–108.9) | 109.6 (107.4–111.8) | 109.0 | 111.2 |
| | Adjusted | 110.5 (109.7–111.3) | 106.6 (104.6–108.7) # | 106.3 | 107.5 |
| DBP | Unadjusted | 75.6 (75.0–76.1) | 75.4 (73.9–76.8) | 74.8 | 76.9 |
| | Adjusted | 76.8 (76.2–77.4) | 73.4 (72.0–74.9) # | 73.0 | 74.5 |

*Reference category.

[1] Smokes cigarettes daily.

[2] Men: average daily number of cigarettes in past week; Women: number of cigarettes in last 24 hours.

# p<0.001.

**Table 4. Mean SBP and DBP: Unadjusted and adjusted for age, BMI, wealth quintile and agricultural occupation.**

| Characteristic | | SBP, mm Hg (95% CI) | | DBP, mm Hg (95% CI) | |
|---|---|---|---|---|---|
| | | Unadjusted | Adjusted | Unadjusted | Adjusted |
| **Men (n = 3420)** | | | | | |
| Age, years | 18–24* | 112.7 (111.5–113.9) | 116.1 (114.4–117.8) | 73.4 (72.3–74.4) | 75.6 (74.3–76.9) |
| | 25–34 | 117.3 (116.4–118.2) [#] | 119.0 (117.6–120.4) [#] | 79.4 (78.5–80.2) [#] | 80.3 (79.2–81.4) [#] |
| | 35–44 | 120.5 (119.2–121.8) [#] | 121.7 (119.9–123.5) [#] | 82.2 (81.2–83.2) [#] | 82.8 (81.5–84.1) [#] |
| | 45 and over | 123.9 (121.4–126.3) [#] | 125.5 (122.9–128.1) [#] | 83.7 (82.2–85.2) [#] | 84.9 (83.2–86.5) [#] |
| BMI, kg/m² | <18.5 | 110.2 (108.7–111.6) [#] | 112.0 (110.2–113.7) [#] | 73.4 (72.2–74.4) [#] | 74.8 (73.6–76.0) [#] |
| | ≥18.5–25* | 116.9 (116.0–117.8) | 118.0 (116.8–119.2) | 78.0 (77.2–78.8) | 78.6 (77.7–79.5) |
| | ≥25–30 | 124.3 (123.0–125.6) [#] | 124.6 (123.1–126.1) [#] | 85.3 (84.2–86.4) [#] | 84.6 (83.4–85.8) [#] |
| | ≥30 | 128.3 (124.2–132.4) [#] | 127.7 (123.4–132.0) [#] | 86.9 (83.9–90.0) [#] | 85.4 (82.3–88.6) [#] |
| Smoking | Current non-smoker*,¹ | 117.3 (116.4–118.2) | 120.2 (118.9–121.5) | 78.8 (78.0–79.5) | 81.0 (80.0–82.0) |
| | Current smoker (manufactured or hand-rolled) | 118.8 (117.0–120.5) | 121.0 (119.0–123.0) | 79.3 (78.0–80.6) | 80.7 (79.4–82.1) |
| **Women (n = 5518)** | | | | | |
| Age, years | 18–24* | 103.3 (102.6–104.0) | 105.1 (103.8–106.4) | 71.7 (71.1–72.3) | 73.2 (72.2–74.2) |
| | 25–34 | 106.5 (105.8–107.3) [#] | 107.2 (105.8–108.5) [#] | 75.1 (74.5–75.8) [#] | 75.6 (74.6–76.5) [#] |
| | 35–44 | 113.2 (112.1–114.3) [#] | 113.2 (111.8–114.6) [#] | 78.8 (78.0–79.6) [#] | 78.8 (77.8–79.8) [#] |
| | 45 and over | 117.9 (116.0–119.8) [#] | 118.2 (116.3–120.1) [#] | 80.6 (79.5–81.8) [#] | 80.9 (79.7–82.1) [#] |
| BMI, kg/m² | <18.5 | 102.9 (101.9–104.0) [#] | 104.0 (102.6–105.4) [#] | 71.2 (70.4–72.0) [#] | 71.6 (70.6–72.6) [#] |
| | ≥18.5–25* | 107.2 (106.6–107.9) | 108.1 (106.9–109.2) | 74.5 (74.0–75.0) | 74.6 (73.8–75.4) |
| | ≥25–30 | 112.8 (111.4–114.3) [#] | 112.9 (111.3–114.5) [#] | 79.9 (78.7–81.0) [#] | 79.3 (78.1–80.6) [#] |
| | ≥30 | 119.3 (117.2–121.4) [#] | 118.7 (116.6–120.9) [#] | 84.0 (82.7–85.4) [#] | 83.0 (81.4–84.5) [#] |
| Smoking | Current non-smoker*,¹ | 108.3 (107.6–108.9) | 112.3 (111.4–113.2) | 75.6 (75.0–76.1) | 78.2 (77.6–78.8) |
| | Current smoker (manufactured or hand-rolled) | 109.6 (107.4–111.8) | 109.6 (107.5–111.6) [¥] | 75.4 (73.9–76.8) | 76.0 (74.6–77.4) [‡] |

¹ Does not smoke cigarettes daily.

² Men: average daily number of cigarettes in past week; Women: number of cigarettes in last 24 hours.

*Reference category.

¥ $p < 0.05$.

‡ $p < 0.005$.

# $p < 0.001$.

Mean BP levels after adjustment for age, BMI, wealth quintile (socio-economic status) and agricultural occupation (proxy for physical activity) are shown stratified by age, BMI and smoking status in Table 4. In both men and women, SBP and DBP increased significantly with increasing age and BMI categories. In men, the mean increase in BP in overweight compared to normal weight was 6–7 mm, and 7–10 mm in obese. In women, the mean increase in BP in overweight compared to normal weight was approximately 5 mm, and 8–10 mm in obese. Women smokers had significantly lower SBP (mean difference 2.8 mm, 95% CI 0.7–4.8 mm) and lower DBP (mean difference 2.2 mm, 95% CI 0.9–3.6 mm) than non-smokers after adjustment. There were no significant differences in BP between smokers and non-smokers in males, either before or after adjustment. Tests for interaction between BMI and the smoking-SBP relationship were not significant in men or women.

## Discussion

This study used data from a nationally representative survey to examine the relationship between daily cigarette smoking and BP in Nepali adults aged 18–49 years. After adjustment for age, BMI and physical activity, no positive association was observed between cigarettes

smoking and BP for men or for women. Indeed, women who smoked cigarettes had significantly lower BP than non-smoking women, by, on average, 2.8 mm and 2.2 mm for SBP and DBP respectively.

While smoking is a major risk factor for cardiovascular disease the association with increased BP is still unclear [34]. Our findings for men and women are consistent with many epidemiological studies showing that BP is either lower or the same in smokers as in non-smokers, the average difference being about 2–8 mm Hg for systolic pressure and 1–5 mm Hg for diastolic [35]. Globally, national surveys and longitudinal studies over the last twenty years have found different patterns for the smoking association with BP related to geography, sex and race. The Health Survey for England [17] found no significant difference in BP of male smokers aged 16–44 years compared to non-smokers, but in other national studies of men, notably in China and Japan, lower BP was found [20, 36]. A longitudinal study in the U.S. with a 30-year follow-up did not find a significant increase in SBP or DBP over time in male or female smokers, and white women smokers had lower DBP [34]. Studies in the UK, Sweden, Israel, and China [17, 22, 37, 38] have also found a lower BP in women current smokers compared to non-smokers, consistent with our findings, and a longitudinal study risk of incident hypertension in women conducted in the U.S. found cigarette smoking does not significantly increase the risk of incident hypertension in women smoking up to 15 cigarettes per day [15]. A meta-analysis of over 20 population-based studies conducted in many geographic locations concluded that there was no causal association between smoking heaviness in current smokers of either sex and SBP or DBP [39].

A systematic review of hypertension in low- and middle-income countries found geographic differences in the relationship between smoking and BP with lower proportions of hypertension among smokers compared to non-smokers in Europe and Central Asia, Latin America and Caribbean, and Middle East and North Africa regions, but higher proportions of hypertension amongst smokers compared to non-smokers in East Asia and Pacific, South Asia, and Sub-Saharan Africa regions [40]. Individual studies in Nepal have produced inconsistent findings, and many have not included adjustments for age and BMI [41–47]. One study in periurban Kathmandu found an association between current cigarette smoking and higher BP after adjustment [18]. Other studies conducted in semi-urban and rural settings in Nepal [48–50], and a nationwide survey [51], found current smoking was not significantly associated with BP in multivariable analyses. A systematic review and meta-analysis of 12 studies undertaken in Nepal in the last 20 years [52] estimated smokers have 1.43 times the odds (95% CI 1.14–1.79) of hypertension based on unadjusted odds ratios which did not control for confounders. Other recent systematic reviews and meta-analyses of the prevalence of hypertension in Nepal [53, 54] did not examine the effect of cigarette smoking. Two studies using data from the same 2016 NDHS survey as the present study to examine risk factors for hypertension [55, 56] used cigarette smoking in the 30 minutes prior to the BP measurement (Y/N) as their smoking variable and hence measured the short-term impact of nicotine on elevating BP [11], rather than the chronic effect of cigarette smoking on BP. The mixed results between studies demonstrate that methodological differences, different populations, and additional unmeasured confounders (e.g., lifestyle, diet, cultural characteristics, physical activity) may influence the observed relationship between smoking and BP.

The strength of our study compared to other studies conducted in Nepal is that we have used data from a nationally representative survey, rather than a specific geographical location, to examine the association between current cigarette smoking and BP. In addition, we used smoking variables that reflect daily smoking patterns rather than smoking in the 30 minutes before BP measurement. We adjusted for age, BMI, socioeconomic status and social differences, but we were unable to adjust for physical activity directly, dietary intake of fruit,

vegetables and salt, or alcohol consumption, as these potential confounders were not collected in the 2016 NDHS. To account for physical activity, we used a proxy variable which provided a limited adjustment for this variable. Alcohol intake is strongly associated with smoking in Nepal [49] and may affect the smoking-BP relationship. However, in the study of Primatesta et al. [17] alcohol consumption did not alter smoking effects on BP for men or women. We also did not exclude individuals who had smoked or consumed alcohol or caffeine within 30 minutes before the BP readings, or who were users of smokeless tobacco, which may be a source of confounding. Other limitations include a lack of statistical power to undertake sub-group analysis by level of smoking. The use of office BP measurements in the NDHS rather than 24-hour ambulatory BP did not enable detection of BP changes throughout the day, including patterns such as the white-coat effect and masked hypertension [57]. In addition, cigarette smoking might act preferentially on central BP, rather than brachial BP, in the development of hypertensive target organ disease [12]. Since this is a cross-sectional study, we could not establish causality due to the lack of temporal relationship between smoking and BP.

Overall, we showed that BP is either lower or the same in daily cigarette smokers as in non-smokers in the age group 18–49 years. Our finding that women cigarette smokers, but not men smokers, had a significantly lower BP than non-smokers may reflect either, (1) the physiological effect of cigarette smoking *per se* in women, or (2) unmeasured dietary, lifestyle or health factors associated with low education, poverty, and agricultural work that were characteristics of the women smokers in the study and which may independently lower BP. Possible unmeasured factors include the consumption of a traditional diet (i.e., high in fiber and vegetables, low in fat) and less sedentary behaviour as part of a more traditional lifestyle [58]. While smoking was associated with a lower BMI in women, BMI was adjusted for in the final model.

## Conclusions

This study describes the association between cigarette smoking and blood pressure in adults in Nepal aged 18–49 years. Our finding that daily cigarette smoking was not associated with increased BP in men or women in this population contributes to the understanding of the relationship between these two major risk factors for cardiovascular disease in populations that share characteristics with Nepal. However, the results of this study should not be used to influence the public health campaigns on smoking cessation as cigarette smoking is a strong independent risk factor for cardiovascular disease. Future research employing longitudinal studies, the use of ambulatory BP monitoring, adjusting for additional confounders, and studies on arterial stiffness and central blood pressure [12] may provide further insight into of the effect of smoking on BP.

## Acknowledgments

We acknowledge the contribution of the Demographic and Health Survey (DHS) program for providing access to the 2016 Nepal dataset.

## Author Contributions

**Conceptualization:** Renqiao Lan, Prakash Dev Pant, Hilary Jane Wallace.

**Data curation:** Renqiao Lan, Prakash Dev Pant, Hilary Jane Wallace.

**Formal analysis:** Renqiao Lan, Max K. Bulsara, Hilary Jane Wallace.

**Funding acquisition:** Hilary Jane Wallace.

**Methodology:** Renqiao Lan, Max K. Bulsara, Prakash Dev Pant, Hilary Jane Wallace.

**Project administration:** Hilary Jane Wallace.

**Resources:** Prakash Dev Pant, Hilary Jane Wallace.

**Software:** Max K. Bulsara.

**Supervision:** Max K. Bulsara, Prakash Dev Pant, Hilary Jane Wallace.

**Validation:** Max K. Bulsara, Hilary Jane Wallace.

**Writing – original draft:** Renqiao Lan, Hilary Jane Wallace.

**Writing – review & editing:** Renqiao Lan, Max K. Bulsara, Prakash Dev Pant, Hilary Jane Wallace.

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
