## [Decision Letter · Decision Letter 0]

6 Sep 2021

 PGPH-D-21-00510 Relationship between cigarette smoking and blood pressure in adults in Nepal: A population-based cross-sectional study PLOS Global Public Health

Dear Dr. %Wallace%,

Thank you for submitting your manuscript to PLOS Global Public Health. After careful consideration, we feel that it has merit but does not fully meet PLOS Global Public Health’s publication criteria as it currently stands. Therefore, we invite you to submit a revised version of the manuscript that addresses the points raised during the review process.

 Dear Authors Please revised the abstract and methods section based on the reviewers comments. 

We look forward to receiving your revised manuscript.

Kind regards,

Palash Chandra Banik, MPhil

Academic Editor

Journal Requirements:

Additional Editor Comments (if provided):

Dear Authors

This manuscript is suitable for the publication after minor changes suggested by the reviewers. I would like to request you for address the reviewer comments and revise the manuscript.

Reviewers' comments:

Reviewer's Responses to Questions

**Comments to the Author**

1. Does this manuscript meet PLOS Global Public Health’s publication criteria? Is the manuscript technically sound, and do the data support the conclusions? The manuscript must describe methodologically and ethically rigorous research with conclusions that are appropriately drawn based on the data presented.

Reviewer #1: Yes

Reviewer #2: Yes

2. Has the statistical analysis been performed appropriately and rigorously?

Reviewer #1: Yes

Reviewer #2: I don't know

3. Have the authors made all data underlying the findings in their manuscript fully available (please refer to the Data Availability Statement at the start of the manuscript PDF file)?

Reviewer #1: Yes

Reviewer #2: Yes

4. Is the manuscript presented in an intelligible fashion and written in standard English?

Reviewer #1: Yes

Reviewer #2: Yes

5. Review Comments to the Author

Reviewer #1: The authors studied the association between cigarette smoking and blood pressure in adults in Nepal aged 18-49 years from a nationally representative survey. The finding that cigarette smoking was not associated with increased BP in Nepal will contribute to the understanding of the relationship between these two major risk factors. Overall, the results are clear, and the data well understood. In my opinion, the manuscript is suitable for publication in "PLOS Global Public Health".

Reviewer #2: This type of study is so important as specially it is using nationally representative data. But on other sense, such negative findings and their expression on public platform can create different views on health perspective. So, editors might think on this expect.

All through the study is beautifully written and almost in a clear sense. Except the Abstract , table composition and conclusion part, the study is so well balanced. The authors should get a heartiest facilitation from my side. But few comments are here which need to be clarified to make the study far better for readers.

ABSTRACT:

Commencement of the abstract was good. But conclusion is too hazy and in some points i failed to understand what the authors actually intended to say. I think a conclusion should be ln such pattern what the readers can adopt in a single reading. So, i will request the authors to work on ABSTREACT and to make it more succinate.

BODY of the STUDY:

Introduction and Methods are well written and clear.

Few queries-

1. Is ciigerrete smoking is so common in women of Nepal? If not, then the title should be corrected using tobacco smoking , rather than cigerrete smoking. It is my opinion, authors can differ and defend.

2. BP measurement: Three times BP measurements have been taken within 5 minutes interval. Does it create any variation? And have you found any reference or guideline for measuring BP in 5 minutes3. interval? If yes, then please mention and clarify your thoughts.

3. Why does the authors keep differences on number of cigerrete smoking for men and women? What is the exact reason here? If any specific reason with reference is available, please mention.

5. Have the authors included SLT in their studies? Again this tern cigerrete smoking is creating a confusion here. But i found , no where it is mentioned properly. Please clarify this.

6. As proper physical activity measurement guidelines, questionnaires are available , so why you considered agriculture activity only as a proxy of physical activity? Is there any specific reason.

Lastly discussion is also well written. But again problem faced to understand the conclusion of the study. Somehow, it is not that clear as like the whole study. So, i wish authors will work on that if they feel.

6. PLOS authors have the option to publish the peer review history of their article (what does this mean?). If published, this will include your full peer review and any attached files.

**Do you want your identity to be public for this peer review?** For information about this choice, including consent withdrawal, please see our Privacy Policy.

Reviewer #1: No

Reviewer #2: **Yes: **Dr. Fardina Rahman Omi, MBBS, MPH in NCD

---

## [Decision Letter · Decision Letter 1]

18 Oct 2021

Relationship between cigarette smoking and blood pressure in adults in Nepal: A population-based cross-sectional study

PGPH-D-21-00510R1

Dear Dr. Wallace,

We're pleased to inform you that your manuscript has been judged scientifically suitable for publication and will be formally accepted for publication once it meets all outstanding technical requirements.

Within one week, you'll receive an e-mail detailing the required amendments. When these have been addressed, you'll receive a formal acceptance letter and your manuscript will be scheduled for publication.

An invoice for payment will follow shortly after the formal acceptance. To ensure an efficient process, please log into Editorial Manager at https://www.editorialmanager.com/pgph/ click the 'Update My Information' link at the top of the page, and double check that your user information is up-to-date. If you have any billing related questions, please contact our Author Billing department directly at authorbilling@plos.org.

Kind regards,

Palash Chandra Banik, MPhil

Academic Editor

Additional Editor Comments (optional):

Dear Authors, Please follow the review comments if any and address accordingly.

Reviewers' comments:

Reviewer's Responses to Questions

**Comments to the Author**

1. If the authors have adequately addressed your comments raised in a previous round of review and you feel that this manuscript is now acceptable for publication, you may indicate that here to bypass the “Comments to the Author” section, enter your conflict of interest statement in the “Confidential to Editor” section, and submit your "Accept" recommendation.

Reviewer #1: All comments have been addressed

Reviewer #2: All comments have been addressed

2. Does this manuscript meet PLOS Global Public Health’s publication criteria? Is the manuscript technically sound, and do the data support the conclusions? The manuscript must describe methodologically and ethically rigorous research with conclusions that are appropriately drawn based on the data presented.

Reviewer #1: Yes

Reviewer #2: Yes

3. Has the statistical analysis been performed appropriately and rigorously?

Reviewer #1: Yes

Reviewer #2: Yes

4. Have the authors made all data underlying the findings in their manuscript fully available (please refer to the Data Availability Statement at the start of the manuscript PDF file)?

Reviewer #1: Yes

Reviewer #2: Yes

5. Is the manuscript presented in an intelligible fashion and written in standard English?

Reviewer #1: Yes

Reviewer #2: Yes

6. Review Comments to the Author

Reviewer #1: (No Response)

Reviewer #2: Authors hard work is praiseworthy. This study will help in evidence generation too.

7. PLOS authors have the option to publish the peer review history of their article (what does this mean?). If published, this will include your full peer review and any attached files.

**Do you want your identity to be public for this peer review?** For information about this choice, including consent withdrawal, please see our Privacy Policy.

Reviewer #1: No

Reviewer #2: No
